# Fermentation of Peanut Slurry with *Lactococcus lactis* Species, *Leuconostoc* and *Propionibacterium freudenreichii* subsp. *globosum* Enhanced Protein Digestibility

**DOI:** 10.3390/foods12183447

**Published:** 2023-09-15

**Authors:** Ayana Saizen, Letitia Stipkovits, Yukiyo Muto, Luca Serventi

**Affiliations:** Faculty of Agriculture and Life Sciences, Lincoln University, Lincoln 7647, New Zealand

**Keywords:** protein digestibility, peanuts, legumes, probiotics, fermented food, *Lactococcus*, *Propionibacterium*, cheese, amino acids, texture

## Abstract

Peanuts contain nutritionally relevant levels of protein, yet are poorly digestible. Fermentation is a promising technique to boost legume protein quality, but its effect on the protein quality of raw peanuts has not been investigated. This study aimed to assess the impact of fermentation on the in vitro protein digestibility and free amino acid profile of cooked peanut slurry (peanut to water ratio 1:1). Cultures used were *Propionibacterium freudenreichii* subsp. *globosum* and a commercial fresh cheese culture that contained *Lactococcus lactis* subsp. *cremoris*, *lactis*, *lactis biovar diacetylactis*, and *Leuconostoc*, fermenting at 38 °C for 48 h. Samples fermented with the combination of cultures showed higher protein digestibility, as well as softer texture. Significant increases were observed only in the sample fermented with the fresh cheese culture. While the fresh cheese culture improved the free amino acid profile after fermentation, the combination of the cultures decreased all free amino acid concentrations except for glutamine, alanine, and proline. The observed increases in in vitro protein digestibility and the free amino acid profile may be attributed to the proteolytic activities of the cultures.

## 1. Introduction

Legumes are one of the main sources of plant-based protein. They are rich in protein, fiber, carbohydrates, and micronutrients [1]. For instance, commercial raw peanuts contain 26.3 g of protein, 11.4 g of carbohydrates, and 5.7 g of dietary fiber per 100 g [2]. While legumes are nutritious foods, plant-based proteins generally have a lower digestibility rate (75 to 80%) compared to animal proteins (90 to 95%) [3]. Additionally, most plant-based proteins are incomplete sources of amino acids, while animal-based proteins tend to be complete sources [4]. In a previous study, it is reported that raw peanuts contained 270 ± 4 g/kg dry weight of protein which mainly consisted of glutamic acid (859 ± 23 mg/kg dry weight), phenylalanine (555 ± 84 mg/kg dry weight), asparagine (238 ± 79 mg/kg dry weight), valine (208 ± 9 mg/kg dry weight), aspartic acid (190 ± 8 mg/kg dry weight), arginine (188 ± 18 mg/kg dry weight), and alanine (170 ± 20 mg/kg dry weight) [5].

The protein quality of legumes can be enhanced through the fermentation technique. The quality of dietary protein is assessed based on its digestibility and amino acid profile [6]. Protein digestibility refers to the amount of hydrolyzed proteins that are broken down by digestive enzymes in relation to the total protein content [3]. Small peptide fractions can contribute to an increase in protein digestibility and accessibility in the body [7]. On the other hand, amino acid profiles can be expressed as free amino acid or essential amino acid profiles [3,8]. Ketnawa and Ogawa [8] discovered that soybeans fermented with *Bacillus natto* increased the protein digestibility with low molecular weight soluble-protein fractions. They also found that the fermentation technique increased the concentration of all free amino acids.

Although fermentation is a promising technique to boost legume protein quality, there is a lack of understanding of how it impacts the protein quality of raw peanuts. It was reported that the incorporation of naturally fermented peanuts into pearl millet-based infant foods improved the in vitro protein digestibility compared to the incorporation of naturally fermented cowpea [9]. It is also reported that sorghum base tempeh fermented with *Rhizopus oligosporus* increased the in vitro protein digestibility when mungbean and peanuts were incorporated [10]. However, there is no published research on the effect of fermentation with cheese cultures on the protein digestibility and free amino acid profile of raw peanuts. The aim of this research was to evaluate the effect of fermentation using a commercial fresh cheese culture and *Propionibacterium freudenreichii* subsp. *globosum* on the protein digestibility and free amino acid profile of raw peanut slurry. The fresh cheese culture contained *Lactococcus lactis* subspecies. The hypothesis posited that proteolysis activities of microbes would increase the protein digestibility and free amino acid profile of raw peanut slurry through fermentation.

## 2. Materials and Methods

### 2.1. Materials and Slurry Fermentation

Raw peanuts (Pams, Auckland, New Zealand) were deskinned and blended with water at a 1:1 (*w*/*w*) ratio until a smooth slurry formed. Fresh cheese culture was obtained from Mad Millie (Auckland, New Zealand); it contained *Lactococcus lactis* subsp. *cremoris*, *lactis*, *lactis biovar diacetylactis*, and *Leuconostoc*. *Propionibacterium freudenreichii* subsp. *globosum* culture was obtained from the Urban Cheese Company (West Melton, Christchurch, New Zealand). The fresh cheese culture was dissolved in water at a concentration of 0.10%. *Propionibacterium freudenreichii* subsp. *globosum* was also dissolved into the fresh cheese culture solution at a concentration of 0.03%. A baking pan was sanitized using 70% ethanol, and then 90 g of the raw peanut slurry was transferred to a cup on the pan. Subsequently, 32 g of each culture solution was added to the cup and mixed in. At this point, pre-fermentation samples were stored in a freezer for subsequent analysis. The samples were incubated at 38 °C for 48 h, then stored in a freezer for further analysis and thawed at room temperature prior to analysis. Each sample was prepared in triplicate. A representative picture of the slurry produced is available in Figure 1.

### 2.2. pH

One of the parameters used to test fermentability is pH. The pH of samples was measured with a Mettler Toledo pH meter (SevenEasy pH, Schwerzenbach, Switzerland). Slurries were analyzed in their original state by immersing the pH probe in them.

### 2.3. Microbial Enumeration

Fermentability was also tested by quantifying *Lactococcus* microorganisms. Duplicate sets of De Man–Rogosa–Sharpe (MRS) agar plates, pH 5.7, were labeled and inoculated with 1 mL of diluted suspensions, distributed with a spreader. The MRS media was chosen as ideal substrate for probiotic growth, specifically for lactobacilli such as *Lactococcus* and *Leuconostoc* [11]. Incubation took place at 35 °C for 48 h. Anaerobic sachets (Oxoid AnaeroGen 2.5 L, Thermo Scientific^TM^; Christchurch, New Zealand) were used. Microbial count was expressed as colony forming units per gram of sample (CFU/g).

### 2.4. Protein Digestibility

#### 2.4.1. Preparation of Sample Solutions

Aliquots of 1 g of raw peanut slurry were weighed and left overnight at room temperature in a 200 mL beaker. Then, 90 mL of distilled water was added to the 200 mL beaker to dissolve the slurry, and the mixture was stirred using a magnetic stirring bar until the slurry broke down into small particles. The pH of the solution was adjusted to 2.0 with 1 mol/L HCl, and then the volume was adjusted to 100 mL with distilled water, for a final pH of 2.0.

#### 2.4.2. Preparation of Enzymes

Pepsin solution was prepared with 25 mg of pepsin (600–800 U/mg protein; EC 232-623-3) and 10 mL of 0.04 mol/L HCl for gastric digestion. Pancreatin solution was prepared with 25 mg of pancreatin (activity > 25 USP units/mg; CAS 8049-47-6) and 10 mL of 0.1 mol/L phosphate buffer for intestinal digestion combined with lipase (activity > 2.0 USP units/mg) and protease (activity > 25 USP units/mg).

#### 2.4.3. In Vitro Protein Digestion

In vitro digestion was determined according to the methods of Uraipong and Zhao [12] with slight adjustments. Sample solutions and blank water were heated to 37 °C using magnetic stirrers (IKA, Guangzhou, China). Then, 1 mL aliquots were taken, followed by the addition of 0.5 mL of the pepsin solution to initiate the gastric phase. Samples were kept at 37 °C and stirred, and 1 mL of aliquots were taken every 20 min up to 120 min. The aliquots were heated in a 95 °C water bath for 10 min to quench the enzymatic reaction at each sampling point. After 120 min, the pH was adjusted to 8.0 with 1.7 mol/L NaOH and 1 mol/L HCl. The addition of 0.5 mL pancreatin solution initiated the intestinal phase, while the samples continued to be stirred and maintained at 37 °C. The 1 mL aliquots were again taken every 20 min for the subsequent 120 min, and the aliquots were treated as in the gastric phase to quench the enzymatic reaction at each sampling point. All in vitro digestion samples were stored at −20 °C for further analysis.

#### 2.4.4. In-Vitro Protein Digestibility Evaluation

Protein concentration in the in vitro digestion samples was determined by the Bradford method. BSA solution (0.5 mg/mL) was diluted to form a standard curve (0, 0.125, 0.25, 0.4, and 0.5 mg/mL) using distilled water. Bio-Rad protein assay dye reagent was diluted at a reagent and water ratio of 1:4 (*v*/*v*), and then the diluted reagent was filtered. Then, 10 μL of each in vitro digestion sample and 200 μL of the diluted dye reagent were added to a cell on 96-well plates and then mixed three times using pipetting. The absorbance of the plates was measured at 595 nm within 1 h after mixing the first sample. As shown in Equation (1), the in vitro protein digestibility was calculated using the equation established by Almeida and collaborators [13]. In this equation, *Ph* represents the protein concentration in the in vitro digestion sample, *Pb* is the protein concentration in the blank, and *P*0 represents the protein concentration in the in vitro digestion sample at 0 min.
(1)$$Protein\;Digestibility\left(\%\right)=(1-\frac{Ph-Pb}{P0})\times 100$$

### 2.5. Total Amino Acids

Wet samples were defatted with the Soxhlet method. Defatted samples were then treated prior to chromatographic analysis. The amino acid profile of the freeze-dried sample was determined upon acid hydrolysis (5.0 mL 6 N HCl solutions heated at 110 °C for 20 h) and subsequent chromatographic analysis (Agilent 1100 Series HPLC system; Santa Clara, CA, USA). The HPLC settings were as follows: 150 × 4.6 mm, C18, 3u ACE-111-1546 column (Winlab, Harborough, UK) at a temperature of 40 °C. Two solvents were used: A (0.01 M disodium phosphate in 0.8% tetrahydrofuran, pH 7.5) and B (50% methanol, 50% acetonitrile) (LiChrosolv Reag, VWR, Radnor, PA, USA). The flow rate was 0.7 mL/min with solvent B increasing from 0 to 100% in 24 min, then decreasing to 0% in 12 min. Derivatization was performed with o-phthalaldehyde and 3-Mercaptopropionic acid for primary amino acids and 9-fluorenylmethyl chloroformate for secondary amino acids. Injection volume was 11.0 µL. A fluorescence detector (excitation 335 nm, emission 440 nm) was used. At 22 min, the detector was switched for secondary amino acid (excitation 260 nm, emission 315 nm).

### 2.6. Free Amino Acids

The quantity of free amino acids in wet samples was determined with the same chromatographic method described in Section 2.5, but without acid hydrolysis, as detailed elsewhere [14].

### 2.7. Texture Profile Analysis

Digestibility of protein is also affected by the texture and structure. Therefore, Texture Profile Analysis (TPA) was performed on peanut slurry. A TA.XT Texture Analyser (Stable Micro Systems, Godalming, UK) was used to perform double compression on slurry samples to a compression rate of 40%. Slurry was analyzed in the baking pan and settings were the following: load cell 50 kg, aluminum probe P/25, and test speed 1.7 mm/s. Parameters measured were hardness and adhesiveness.

### 2.8. Statistical Analysis

Values were repeated in duplicate (microbial enumeration, total and free ammino acids) and triplicate (pH, protein digestibility, hardness and adhesiveness). All calculations were performed using Excel from Microsoft Office Home and Business 2019, and Minitab (version 20) was utilized for statistical analysis. Results are reported as mean plus/minus standard deviation. Statistically significant differences were evaluated with one-way analysis of variance (ANOVA) and Tukey’s honest significant difference (HSD) test (*p* < 0.05).

## 3. Results

### 3.1. Fermentability

As shown in Table 1, the pH of the raw peanut slurry decreased significantly after fermentation. The initial pH was 6.78 in the sample with the fresh cheese culture, and 6.67 in the sample with the combination of the fresh cheese culture and *Propionibacterium freudenreichii* subsp. *globosum* (Figure 2). Although the pH significantly decreased to 4.74 and 4.70 after fermentation, no significant difference was observed between the cultures used (Table 1).

Microbial enumeration of *Lactobacillus* species confirmed the fermentability of the peanut slurry samples. Both settings (fresh cheese culture alone, and fresh cheese culture added with *Propionibacterium freudenreichii* subsp. *globosum*) resulted in significant microbial growth for lactobacilli (Table 1).

### 3.2. Protein Digestibility

Figure 2 presents the results of the in vitro protein digestibility. The protein digestibility increased through the simulated gastric and intestinal digestions in all samples (Figure 2). The sample fermented with the combination of the fresh cheese culture and *Propionibacterium freudenreichii* subsp. *globosum* exhibited higher protein digestibility after 100 min than the sample fermented with the fresh cheese culture (Figure 2). Significant differences were observed in both the sample fermented with the fresh cheese culture after 180 min and the sample fermented with cheese culture and *Propionibacterium freudenreichii* subsp. *globosum* after 200 min (Figure 2). No significant difference was observed among different samples at the same digestion time points (Figure 2).

### 3.3. Total Amino Acid Profile

Quantification of total amino acids revealed abundance of glutamic acid (Glu), asparagine (Asn). As expected for legumes, lysine (Lys) was more abundant than methionine (Met). Fermentation with either *Lactococcus* alone or in combination with *Propionibacterium freudenreichii* subsp. *globosum* did not significantly alter the amino acid profile of peanuts (Figure 3).

### 3.4. Free Amino Acid Profile

Figure 4 shows the changes in the free amino acid profile before and after fermentation. The levels of all free amino acids increased after fermentation with the fresh cheese culture (Figure 4). On the other hand, fermentation with the combination of the fresh cheese culture and *Propionibacterium freudenreichii* subsp. *globosum* decreased the amount of all amino acids except for Gln, Ala, and Pro (Figure 4).

### 3.5. Texture

Textural analysis of peanut slurry revealed changes induced by fermentation. Specifically, the cheese culture addition resulted in a drastically softer texture: 52.8 vs. 220 g (Table 2). Softening of the fermented peanut slurry represented a four-fold reduction in hardness. On the contrary, *Propionibacterium freudenreichii* subsp. *globosum* addition did not produce the same results. This result correlated with the changes observed in protein digestibility (Figure 2).

No significant differences were observed in terms of adhesiveness, yet larger variation was observed in the *Propionibacterium freudenreichii* subsp. *globosum*-containing recipe (Table 2).

## 4. Discussion

### 4.1. Fermentability

There was a significant decrease in the pH of the raw peanut slurry after fermentation (Table 1). A possible explanation for this result could be that acids produced by microbes during fermentation resulted in the pH decrease in the samples. Both *Lactococcus lactis* and *Leuconostoc*, which belong to lactic acid bacteria, ferment sugars to produce lactic acid, subsequently lowering the pH [15]. *Propionibacterium* also ferments sugars, producing propionic acid that reduces the pH [16]. Additionally, it was reported that *Propionibacterium freudenreichii* subsp. *globosum* consumed lactate and produced both propionate and acetate within 46 h when incubated at 30 °C [17]. Therefore, the decrease in the pH of the samples can be attributed to lactic acid, propionic acid, and acetate produced by the microbes during fermentation. The sugar profile of peanuts is mostly comprised of a disaccharide (sucrose) and oligosaccharides (stachyose and raffinose) [18]. Whereas sucrose can be fermented by numerous microorganisms, oligosaccharides are efficiently fermented by probiotic microorganisms such as *Lactococcus*. Fermentation of sugars by *Lactococcus lactis* has been shown to produce lactic acid, 2,3-butanediol, and ornithine (an amino acid) [19]. 2,3-butanediol is a known plasticizer used as precursor in the rubber industry [20]. In addition, it delivers a creamy and fruity flavor [21]. The presence of this diol might explain the beneficial effects on protein solubility and texture.

Microbial growth was observed in both samples (Table 1). Peanuts can be an excellent fermentation matrix for probiotic bacteria such as *Lactobacillus* and *Propionibacterium* species. This is due to peanuts’ nutritional value, delivering as much as 24 g of protein per 100 g sample, as well as 4.9 g of sugar [22]. In addition, peanuts contain nutritionally relevant levels of micronutrients such as vitamins B1, B2, B3, B6, B9, and choline [22] and minerals such as calcium, iron, magnesium, phosphorous, potassium, sodium, zinc, copper, manganese, and selenium [23]. These nutrients support microbial growth. Production of a slurry involves grinding peanuts into fine particles. This process can release amino acids, thus providing microorganisms with easily digestible nitrogen [24]. It must be stated that the initial number of probiotics was lower than in previous studies: approximately 3 vs. 6–8 CFU/g [24,25] due to the supplier recommendation for the cultures used. Post-fermentation results were indicated as TMC, as opposed to specific values. The goal of this study was not to quantify the probiotic level, but rather to verify whether fermentation of this matrix is possible.

### 4.2. Protein Digestibility

All samples demonstrated an increase in protein digestibility through the in vitro digestion process (Figure 2). The same trend was observed in a previous study. Ketnawa and Ogawa [8] fermented soybeans with *Bacillus natto* at 40 °C for 18 h. The protein digestibility of soaked, boiled, and fermented soybeans increased significantly as the simulated digestion time increased [8]. Despite the increase in protein digestibility in all samples, there were no significant differences observed in the samples before fermentation. One possible explanation for this might be the non-uniformity of the raw peanut slurry due to the amino acid profile. Raw peanuts mainly consist of both water-soluble and insoluble amino acids. In a previous study, it was revealed that raw peanuts mainly consisted of glutamic acid (859 ± 23 mg/kg dry weight), phenylalanine (555 ± 84 mg/kg dry weight), asparagine (238 ± 79 mg/kg dry weight), and valine (208 ± 9 mg/kg dry weight) [5]. It can be inferred that the insoluble proteins contribute to the non-uniformity of the peanut slurry before fermentation. This study could be improved by agitation before sampling to obtain uniform products.

Significant increases in protein digestibility during intestinal digestion were observed in the raw peanut slurry fermented with the fresh cheese culture after 180 min (Figure 2). The increases in protein digestibility can be attributed to small peptides produced by *Lactococcus lactis* and *Leuconostoc* in the fresh cheese culture, due to the production of proteases. *Lactococcus lactis* are proteolytic microbes that produce proteases during cell growth [25,26]. In a previous study, *Lactococcus lactis* was used as a proteolytic microbe to obtain bioactive peptides from soymilk through fermentation [27]. After fermentation, the soymilk had 2875 U/mL of protease activity, and the degree of hydrolysis value reached 44.32%. The degree of hydrolysis was assessed with the trinitro-benzene-sulfonic acid assay and calculated as the ratio between the increase in amino acid content after hydrolysis and the total peptide content, with the result expressed as a percentage [27]. *Leuconostoc* may also have a proteolytic ability. Rizzello and collaborators [28] fermented 300 g of a fava bean flour and water mixture using 6 log cfu/g of *Leuconostoc kimchi* at 25 °C for 48 h. After the fermentation, the peptide content significantly increased from 16.03 ± 0.94 g/kg to 20.18 ± 1.12 g/kg [28]. Researchers concluded that proteases produced by *Leuconostoc kimchi* hydrolyzed proteins, and the hydrolyzed peptide fractions increased the protein digestibility significantly, which increased from around 55% to 65% during the simulated gastrointestinal digestion. Further studies may explore the optimal temperature to improve the protein digestibility using the fresh cheese culture. Both *Lactococcus lactis* and *Leuconostoc* are mesophilic bacterium. While the optimal growth temperature of *Lactococcus lactis* was around 30 °C, the optimal temperature of *Leuconostoc* for maximum growth was from 34 to 36 °C [29,30]. In this study, the raw peanut slurry was incubated at 38 °C which was higher than the optimal one suggested by previous studies. Further research could assess the optimal temperature to maximize the protein digestibility of the fermented raw peanut slurry using the fresh cheese culture. The moisture content of the slurry was 70% for all samples treated, providing sufficient moisture for the proteolysis to take place.

While the raw peanut slurry fermented with a combination of the fresh cheese culture and *Propionibacterium freudenreichii* subsp. *globosum* demonstrated higher protein digestibility after 100 min than the sample fermented solely with the fresh cheese culture, significant differences were only observed at the 200 min (Figure 2). One potential explanation for this could be the formation of aggregates in the samples containing *Propionibacterium freudenreichii* subsp. *globosum*, contributing to the non-homogeneity of the peanut peptides. The proteinase activity of *Propionibacterium freudenreichii* subsp. *globosum* may be attributed to the following enzymes: aminopeptidase, iminopeptidase, X-prolyl dipeptidyl aminopeptidase, endopeptidase, two different oligopeptidases, and carboxypeptidase [31].

In a previous study, it was found that *Propionibacterium freudenreichii* subsp. *globosum* produced extracellular polysaccharides in a yeast extract–lactate medium and formed aggregates on a yeast extract–lactate plate [32]. The researchers concluded that the produced extracellular polysaccharides contributed to the aggregate characteristics of the *Propionibacterium freudenreichii* subsp. *globosum*. Based on this, it could be inferred that the aggregates formed by *Propionibacterium freudenreichii* subsp. *globosum* producing polysaccharides contributed to the non-homogeneity of the fermented peanut slurry. Further research could examine the protein digestibility of fermented raw peanut slurry using the combination of the fresh cheese culture and *Propionibacterium freudenreichii* subsp. *globosum* after agitation.

### 4.3. Total Amino Acid Profile

The total amount of acids in the slurries ranged from 40.4 to 43.7 µmol, without significant changes due to fermentation. This agreed with the previous literature on peanuts: total amino acids 45 µmol [33]. The most abundant amino acids found in the peanut slurry were Glu and Asp. This result also agreed with the previous literature [32]. Within essential amino acids, lysine (Lys) was three times more abundant than methionine (Met), as expected for legume seeds [30].

### 4.4. Amino Acid Profile

All free amino acid concentrations increased after fermentation using the fresh cheese culture (Figure 4). The increase can be attributed to proteolytic microbes in the fresh cheese culture hydrolyzing proteins in the raw peanut slurry. In a previous study, soybeans were fermented using *Bacillus natto* at 40 °C for 18 h [8]. After the fermentation, all free amino acid concentrations increased except for Arg [8]. The researchers concluded that hydrolyzation during fermentation with the proteolytic microbe improved the free amino acid profile. Although it can be inferred that fermentation improved the free amino acid profile of raw peanut slurry, this research was unable to determine significant differences. Further research could identify the specific amino acids that experience significant increases during fermentation with the fresh cheese culture. Further studies could also examine the changes in free amino acid profile after in vitro digestion to estimate nutritional availability in the body.

On the other hand, fermentation with the combination of the fresh cheese culture and *Propionibacterium freudenreichii* subsp. *globosum* led to a decrease in the concentration of all amino acids, except for Gln, Ala, and Pro. It could be inferred that *Propionibacterium freudenreichii* subsp. *globosum* consumed the free amino acids produced by the fresh cheese culture during fermentation. In a previous study, *Propionibacterium freudenreichii globosum* was inoculated in yeast extract–lactate medium and incubated at 30 °C [34]. The researchers found that the *Propionibacterium freudenreichii* subsp. *globosum* strain consumed all free amino acids except for Thr, Gln, Val, Met, Lie, Try, His, and Pro in yeast extract–lactate medium after 3 days of incubation [32]. Based on this, it could be inferred that *Propionibacterium freudenreichii* subsp. *globosum* might consume the amino acids hydrolyzed by the fresh cheese culture. Aburjaile and coauthors [34] also mentioned that the concentration of several free amino acids increased after 9 days of fermentation. Future studies could investigate the change in free amino acid concentrations over different fermentation periods.

Organoleptic quality is crucial. The current study estimated sensory quality instrumentally: free amino acids and peptides affect aroma and flavor, hardness and adhesiveness affect texture. Free amino acids contribute to the aroma and flavor of food. Fermentation with the cheese culture freed glutamic acid and asparagine (Figure 4). Glutamic acid is known to produce the umami taste [35], as some hydrolysate peanut peptides do [36]. This is a flavor profile that typically entices consumers, as it occurs with cheese, meat, and mushrooms. Interestingly, asparagine flavor profile depends on its structure. While L-asparagine is described as tasteless, D-asparagine was described as having an intense sweet flavor [37]. These changes in the amino acidic profile can potentially increase consumer acceptance of the fermented peanut slurry. In addition, *Propionibacterium freudenreichii* subsp. *globosum* is known to reduce the acidity of fermented food. Therefore, it is possible that fermentation of peanut slurry with *Lactobacillus lactis* might increase consumer acceptability of this highly digestible protein source. No significant changes in pH and fat content were observed among fermented and unfermented samples (Table 1).

### 4.5. Texture

Fermentation using the fresh cheese culture significantly decreased the hardness of the peanut slurry, resulting in a more homogeneous product with a smaller standard deviation (Table 2). The increased protein solubility (Figure 2), which is the second most abundant component after water, likely explains the decrease in the hardness [38,39]. Another factor to consider is pH. Protein structure changes at different pH values. Peanut protein was determined to be more soluble, thus leading to a softer texture, at pH 7 than 5 [40]. Therefore, increased solubility of the acidified slurry (post-fermentation) is remarkable and likely attributed to protein hydrolysis. One possible explanation is that lactic fermentation of legume protein increases its emulsifying properties [41]. The peanut slurry was abundant in oil and water, resulting in an unstable matrix, as shown by the high standard deviation of the raw samples (Table 2). Higher emulsifying ability results in a homogeneous structure and lower surface tension, hence, a softer texture. *Propionibacterium freudenreichii* subsp. *globosum* metabolizes lactic acid into propionic acid, pyruvic acid, carbon dioxide, and vitamin B12 [42]. The main reason for using this microorganism was the synthesis of vitamin B12 in plant-based food.

On the other hand, no significant difference was observed in the sample fermented using both the fresh cheese culture and *Propionibacterium freudenreichii* subsp. *globosum* (Table 2). Only the unfermented sample with both cultures (FCP) was drastically more adhesive than the other three treatments. It has been hypothesized that the propionic culture contained compounds such as complex sugars which might contribute to the sticky texture. The lower standard deviation observed in the fermented slurry might also be attributed to the increased protein solubility. Peanuts are lipid-rich seeds and have low affinity to water, which are composed of approximately 50% oil and 25% protein [23]. Enhanced protein solubility may increase the water affinity of ground peanuts, potentially resulting in a more homogeneous peanut slurry.

On the contrary, adhesiveness was not affected by any treatment. This indicates that soluble protein did not interact with moisture distribution within the slurry. Consequently, it can be stated that the peanut slurry fermented with cheese culture was significantly softer.

## 5. Conclusions

This study has shown that protein digestibility of peanuts can be increased by means of lactic fermentation. Specifically, incubation of a peanut slurry with *Lactobacillus* species and *Leuconostoc* enhanced protein digestibility while decreasing slurry hardness. The slurry fermented with the fresh cheese culture exhibited an increase in protein solubility, from 35 to 50%, after a 240-h in vitro digestion. Simultaneously, slurry hardness decreased from 220 to 53 g upon lactic fermentation, and from 141 to 40 g with *Propionibacterium freudenreichii* subsp. *globosum*.

Probiotic bacteria grew abundantly in this highly nutritious media, releasing amino acids that contribute to umami and sweet taste. Fermentation with *Propionibacterium freudenreichii* subsp. *globosum* did not significantly increase the protein solubility, while it did soften the texture. This study highlighted the benefits of fermenting peanuts with these starter cultures: more digestible protein, sweeter umami taste, and softer texture. Furthermore, *P. globosum* has the potential to synthesize vitamin B12, an essential nutrient lacking in plant-based foods. Potential challenges include fermentation time and cost.

## Figures and Tables

**Figure 1 foods-12-03447-f001:**
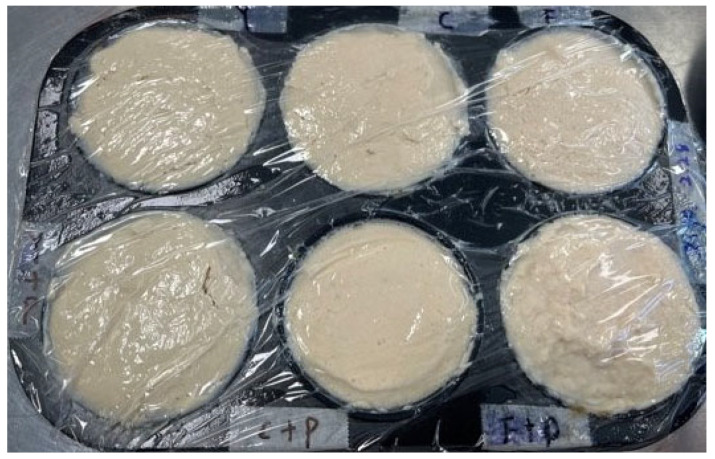
Representative picture of the fermented slurry produced from peanuts.

**Figure 2 foods-12-03447-f002:**
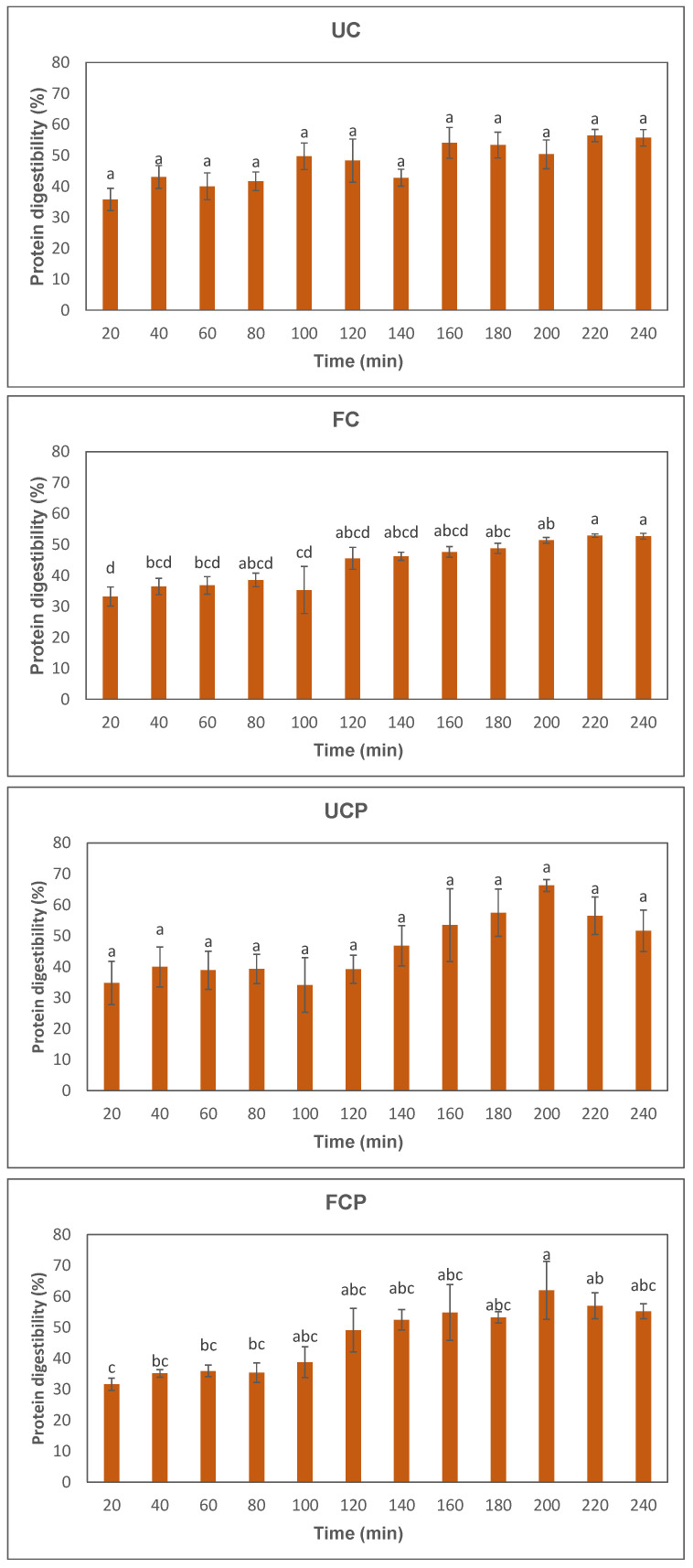
Changes in protein digestibility of peanut slurry. UC = unfermented peanut slurry with the fresh cheese culture; FC = fermented peanut slurry with the fresh cheese culture; UCP = unfermented peanut slurry with the fresh cheese culture and *Propionibacterium freudenreichii* subsp. *Globosum;* FCP = fermented peanuts slurry with the fresh cheese culture *Propionibacterium freudenreichii* subsp. *globosum*. Different letters present a significant difference (α = 0.05) between the same sample, determined by one-way ANOVA and HSD test.

**Figure 3 foods-12-03447-f003:**
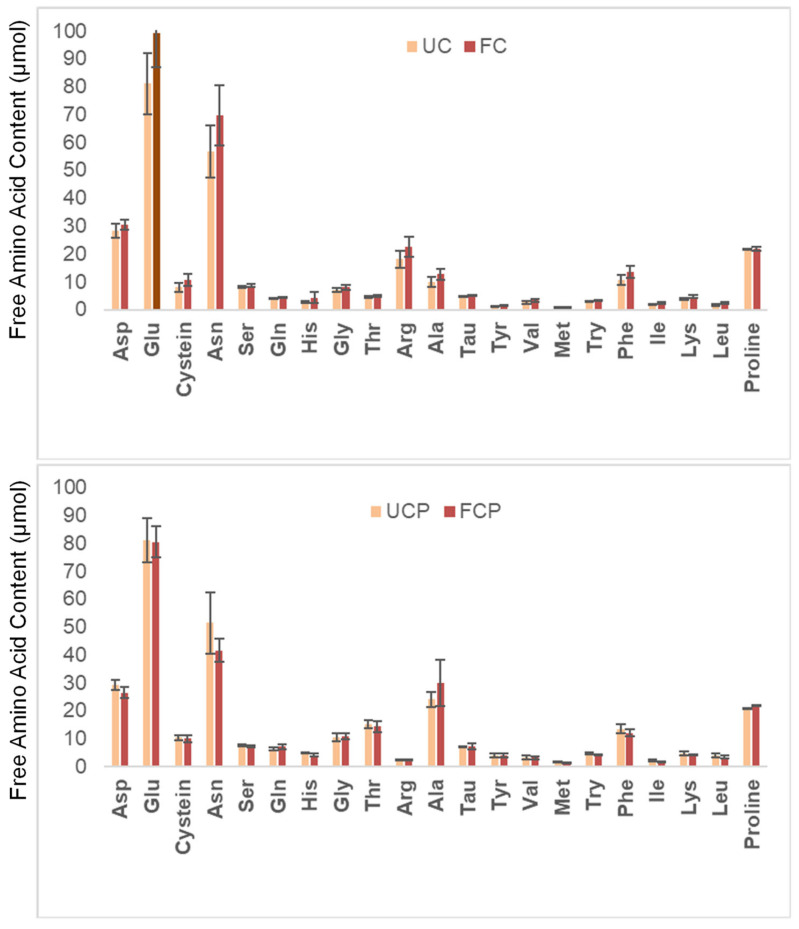
Changes in the total amino acid content. UC = unfermented peanut slurry with the fresh cheese culture; FC = fermented peanut slurry with the fresh cheese culture; UCP = unfermented peanut slurry with the fresh cheese culture and *Propionibacterium freudenreichii* subsp. *Globosum;* FCP = fermented peanuts slurry with the fresh cheese culture *Propionibacterium freudenreichii* subsp. *globosum*. Results are expressed as µmol per dry matter. Statistically significant difference was determined by one-way ANOVA with HSD test (α = 0.05).

**Figure 4 foods-12-03447-f004:**
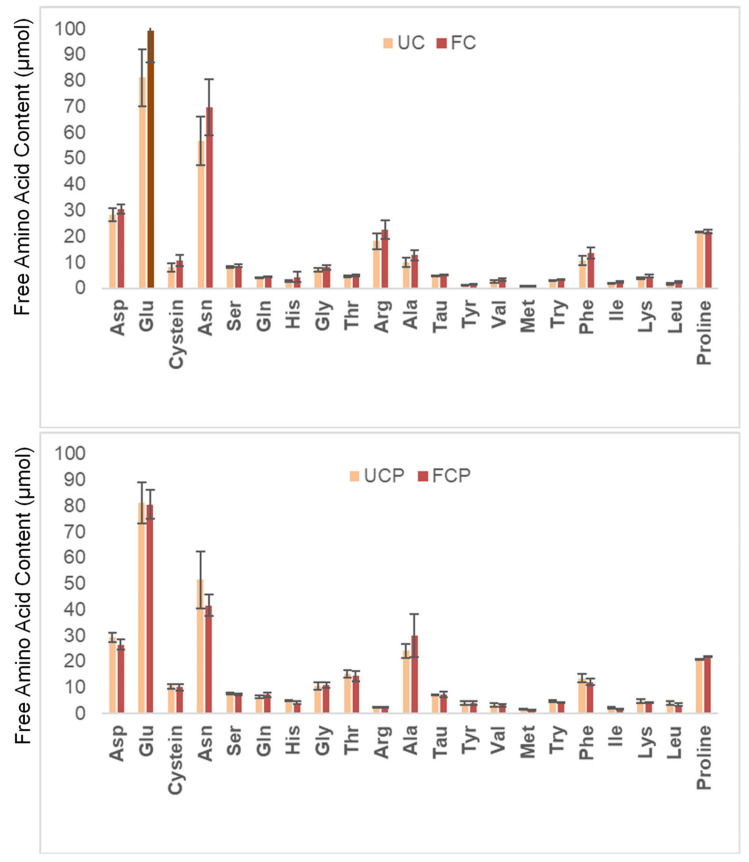
Changes in the free amino acid content. UC = unfermented peanut slurry with the fresh cheese culture; FC = fermented peanut slurry with the fresh cheese culture; UCP = unfermented peanut slurry with the fresh cheese culture and *Propionibacterium freudenreichii* subsp. *Globosum;* FCP = fermented peanuts slurry with the fresh cheese culture *Propionibacterium freudenreichii* subsp. *globosum*. Results are expressed as µmol per dry matter. Statistically significant difference was determined by one-way ANOVA with HSD test (α = 0.05).

**Table 1 foods-12-03447-t001:** Fermentability of peanut slurry measured as pH and microbial enumeration of *Lactobacillus*. Mean values and standard deviation are reported.

Fermentability	UC	FC	UCP	FCP
pH	6.78 ± 0.03 ^a^	4.74 ± 0.06 ^b^	6.67 ± 0.04 ^a^	4.70 ± 0.06 ^b^
Microbial Enumeration (CFU/g)	6 × 10^3^	TMC ^1^	1 × 10^3^	TMC ^1^

UC = unfermented peanut slurry with the fresh cheese culture; FC = fermented peanut slurry with the fresh cheese culture; UCP = unfermented peanut slurry with the fresh cheese culture and *Propionibacterium freudenreichii* subsp. *Globosum;* (FCP) = fermented peanuts slurry with the fresh cheese culture *Propionibacterium freudenreichii* subsp. *globosum*. Different letters present a significant difference (*p* < 0.05) between the same sample, determined by one-way ANOVA and HSD test. ^1^ TMC = Too Many to Count.

**Table 2 foods-12-03447-t002:** Texture profile analysis of the slurry, both unfermented and fermented. Mean values and standard deviation are reported.

Texture Profile	UC	FC	UCP	FCP
Hardness (g)	220 ± 108 ^a^	52.8 ± 4.8 ^b^	141 ± 47 ^ab^	100 ± 3 ^ab^
Adhesiveness (g·mm)	82.9 ± 19.7 ^b^	88.5 ± 16 ^b^	299 ±175 ^a^	173 ± 3 ^ab^

UC = unfermented peanut slurry with the fresh cheese culture; FC = fermented peanut slurry with the fresh cheese culture; UCP = unfermented peanut slurry with the fresh cheese culture and *Propionibacterium freudenreichii* subsp. *Globosum;* (FCP) = fermented peanuts slurry with the fresh cheese culture *Propionibacterium freudenreichii* subsp. *globosum*. Different letters present a significant difference (*p* < 0.05) between the same sample, determined by one-way ANOVA and HSD test.

## Data Availability

The data used to support the findings of this study can be made available by the corresponding author upon request.

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
