# Peer review of "Fermentation of Peanut Slurry with Lactococcus lactis Species, Leuconostoc and Propionibacterium freudenreichii subsp. globosum Enhanced Protein Digestibility"

_foods, 2023, doi:10.3390/foods12183447_

Round 1
Reviewer 1 Report
The manuscript must be improved. The attached file contains the corresponding observations

Author Response
The authors thank Reviewer 1 for the useful feedback.
Attached is our response.

Reviewer 2 Report
The work described in this paper is of interest to the readers of this journal. Nonetheless, there are several issues to be addressed. The manuscript is not acceptable for publication in its present form.
· The whole manuscript: Lactobacillus lactis or Lactococcus lactis?
· Abstract: The text does not contain any information about the use of the species Lactobacillus lactis, while the species Lactococcus lactis subsp cremoris, lactis, lactis biovar diacetylactis are mentioned. I propose to include in the summary those microorganisms that actually participated in experiments and also present important test results. The summary should not contain any abbreviations.
· Keywords: They are not consistent with the manuscript title, abstract, and analytical scope of the research presented. I suggest you choose more appropriate keywords.
· Why wasn't Propionibacterium mentioned in the title and abstract?
· Introduction: In my opinion, the introduction should be expanded to include the importance of LAB and Propionibacterium for the digestibility and bioavailability of legume proteins. There are numerous studies in this area. In addition, it would be valuable to demonstrate the digestibility and bioavailability of peanuts and legumes unfermented. The purpose of the research should be well presented. The introduction and overall manuscript would benefit from using more recent references for general topics.
· 2.2 pH: I propose to detail the preparation of the fermented peanut sample for this analysis.
· 2.3 Microbial Enumeration: Lactobacillus or Lactococcus? Why was MRS used for analysis? What was the pH of this medium? I propose to detail the preparation of the fermented peanut sample for this analysis. Was the final result expressed in 1mL sample or 1g sample? Why could the Propionibacterium number not be determined?
· 2.4.1 Preparation of Sample Solutions: For what purpose was distilled water (of pH 7.0) added to the solution adjusted to pH 2.0? What was the final pH of the sample?
· 2.5 Free Amino Acids: I propose to describe in more detail the preparation of samples for analysis. Merely citing references does not represent the methodology well.
· 2.6 Texture Profile Analysis: How does this relate to the digestibility of peanuts?
· Lines 157-159: The results of the presented microorganism population do not look good. What about the population of Propionibacterium? Why were the cultures performed with such a low sample dilution? The population at 10^3 is insufficient for fermentation samples. Normally, the microorganism population in fermentation reaches 10^6. There are references to this. I propose to discuss this critically in the manuscript.
· 3.2. Protein Digestibility and 3.3. Free Amino Acid Profile: In my opinion, the results should be correlated with the Lactobacillus or Lactococcus and Propionibacterium population. Also, the pH change should correlate in these results.
· 3.4. Texture: What can you explain the SD values in Table 2? I suggest removing "1 TMC = Too Many to Count" because it does not apply in this table. In my opinion, the results should be correlated with the Lactobacillus or Lactococcus and Propionibacterium population. Also, the pH change should correlate in these results.
· Lines 219-227: Does LAB only produce lactic acid? I propose to discuss the composition of the starter microorganisms used, including the homo/hetero fermentation of this LAB. What about Propionibacterium? What sugars are in peanuts and are used by LAB and Propionibacterium for fermentation?
· Lines 229-237: The discussion of the results should be implemented by deepening the findings that may be an advancement of knowledge. There are aspects that are brought as findings of the study that are actually long-established evidence. On what basis do the authors conclude that microbial growth was exponential, since the results received are "Too Many to Count"? Lactobacillus and Propionibacterium species are not probiotic bacteria. How important was the Propionibacterium starter for the samples? What benefits did it bring? I propose to discuss the use of Propionibacterium to ferment plant-based products by other authors. There are references for this. What does the fragment mean: „This is due to peanuts nutritional value, delivering as much as 24 g of protein per 100 g sample, as well as 4.9 g of sugar [16]. In addition, peanuts contain nutritionally relevant levels of micronutrients such as vitamins B1, B2, B3, B6, B9 and choline [16] and minerals calcium, iron, magnesium, phosphorous, potassium, sodium, zinc, copper, manganese, and selenium [17].”?
· Lines 253-288: Mesophilic lactic acid bacteria enzymes work very slowly and are only important for the digestibility of proteins in cheese production during the cheese maturing phase. There are references for this. Proteolytic enzymes from lactic acid bacteria are normally intracellular and are only released after cell death. These enzymes need water, or do you know how high the water content was in the samples? Their importance for protein digestibility is only important with the high lactic acid bacteria population. I propose to discuss it closely using literature data. Likewise, there is much evidence of the use of Propionibacterium for the digestibility of proteins in cheese and plant-based analogues.
· Line 261: Leuconostoc in italics.
· Line 305: P. globosum in italics. Ensure that all names of microorganisms appear in italics throughout the document.
· Lines 315-324: Free amino acids are also a matter of taste. Does peanut protein digestibility matter for the flavor profile of peanuts? What about bitter peptides? They are an important issue in the production of rennet cheese for aging. I suggest deleting this paragraph or expanding it to include a reliable discussion of all consequences of peanut protein digestibility.
· Lines 328-337: The texture of peanuts is not only due to the conversion of proteins, but also of fat. I suggest discussing it. How does starter microorganism selection affect the texture of peanuts? There is ample evidence of the use of lactic acid bacteria and Propionibacterium in cheese and vegetable analogues. The authors should focus on interpreting the results in the context of lactic acid bacteria and Propionibacterium data. What about pH changes?
· 5. Conclusions: I propose to expand this part of the manuscript, taking into account the purpose of the research and the title of the manuscript. What are the potential benefits of the results obtained? What are the limitations of these experiments and results? Both the summary and the conclusions should be supplemented with the specific results of the analyses.
Just some minor spelling errors.
Author Response
The authors thank Reviewer 2 for the useful feedback.
Attached is our response.

Round 2
Reviewer 2 Report
The authors have improved the manuscript and I am partially satisfied with their answers to my questions. Thank you for the activity of this author. Unfortunately, the manuscript contains more information that needs improvement or has appeared. Below are the ones that caught my eye:
· Proper spelling “Propionibacterium freudenreichii subsp. globosum” must be corrected throughout the entire manuscript, including in the title of the article: „Propionibacterium freudenreichii subsp. globosum” or „P. freudenreichii subsp. globosum”. The spelling such as: “Propionibacterium freudenreichii globosum”, “P. globosum”, “P. freudenreichii”, or “Propionibacterium f.” is wrong.
· Line 296: Please correct "Bacillus spp. natto" on "Bacillus natto".
· Please pay attention to the Latin abbreviations used throughout the manuscript: sp. – singular unknown species from a known genus; spp. – more than one unknown species within a known genus; ssp – singular unknown subspecies; subsp – singular unknown subspecies; sspp – plural unknown subspecies; subsp – plural unknown subspecies. Lactococcus, Leuconostoc, and Propionibacterium are three names of bacteria genus. Lactococcus lactis is the name of the bacterial species. Propionibacterium freudenreichii subsp. globosum is the name of the bacterial subspecies. The spelling such as: “Lactococcus lactis spp.” or “P. freudenreichii spp.” is wrong.
· The authors did not explain why they did not examine the population of propionic bacteria and why they used MRS medium to determine the Lactococcus population while M17 medium is recommended for these bacteria. Please answer my doubts.
The manuscript requires thorough reading and improvement, including punctuation.
Author Response
Thanks to Reviewer 2 for the precious feedback.
Our response is attached.

Round 3
Reviewer 2 Report
The authors have improved the manuscript but still the manuscript requires some improvement:
· „Propionibacterium freudenreichii subsp. globosum” or „P. freudenreichii subsp. globosum” - these are the only correct spellings of the name of these bacteria. Please, correct it throughout the entire manuscript, including in the title of the article. The spelling such as: “Propionibacterium freudenreichii globosum”, “P. globosum”, “P. freudenreichii”, or “Propionibacterium f.” is wrong.
· Please write "Bacillus natto" in italics - generic and species names should be written in italics.
Author Response
Response to Reviewer 2, Round 3
The authors have improved the manuscript but still the manuscript requires some improvement:
- „Propionibacterium freudenreichii subsp. globosum” or „P. freudenreichii subsp. globosum” - these are the only correct spellings of the name of these bacteria. Please, correct it throughout the entire manuscript, including in the title of the article. The spelling such as: “Propionibacterium freudenreichii globosum”, “P. globosum”, “P. freudenreichii”, or “Propionibacterium f.” is wrong.
- Please write "Bacillus natto" in italics - generic and species names should be written in italics.
Thank you for noticing these mistakes. The spelling of the Propionibacterium was uniformed to the full name: Propionibacterium freudenreichii subsp. globosum. Bacillus natto was written in Italic throughout the manuscript.
